# Relationship Between Brain Insulin Resistance, Carbohydrate Consumption, and Protein Carbonyls, and the Link Between Peripheral Insulin Resistance, Fat Consumption, and Malondialdehyde

**DOI:** 10.3390/biomedicines13020404

**Published:** 2025-02-07

**Authors:** Elena Salazar-Hernández, Oscar Ezequiel Bahena-Cuevas, Juan Miguel Mendoza-Bello, Martha Isela Barragán-Bonilla, Manuel Sánchez-Alavez, Mónica Espinoza-Rojo

**Affiliations:** 1Laboratory of Molecular Biology and Genomic, Faculty of Biological Chemical Sciences, Autonomous University of Guerrero, Chilpancingo 39090, Guerrero, Mexico; ele_sahe@hotmail.com (E.S.-H.); obahenacuevas@gmail.com (O.E.B.-C.); jmiguelmb29@gmail.com (J.M.M.-B.); martha_isela90@hotmail.com (M.I.B.-B.); 2Faculty of Medicine and Psychology, Autonomous University of Baja California, Tijuana 22390, Baja California, Mexico; manuel.sanchez.alavez@uabc.edu.mx

**Keywords:** diet, insulin resistance, oxidative damage

## Abstract

The consumption of a high-fat (HFD) or high-carbohydrate/low-fat (LFD) diet is related to insulin resistance; however, central and peripheral alterations can occur independently. In this study, the timeline of insulin resistance was determined while taking into consideration the role of diet in oxidative damage. **Background/Objectives**: The aim of this study was to ascertain whether a HFD or LFD induces peripheral insulin resistance (PIR) before brain insulin resistance (BIR), and whether the timing of these alterations correlates with heightened oxidative damage markers in plasma, adipose tissue, and the cerebral cortex. **Methodology and Results**: Three-month-old C57BL/6 male mice were fed with a HFD, LFD, or standard diet for 1, 2, or 3 months. Glucose and insulin tolerance tests were performed to determine PIR, and the hypothalamic thermogenic response to insulin was used to determine their BIR status. For oxidative damage, the levels of malondialdehyde (MDA) and the protein carbonyl group (PCO) and the enzymatic activity of glutathione peroxidase (GSH-Px) were evaluated in plasma, white adipose tissue, brown adipose tissue, and the cerebral cortex. PIR occurred at 3 months of the HFD, but MDA levels in the white adipose tissue increased at 2 months. BIR occurred at 1 and 2 months of the LFD, but the enzymatic activity of GSH-Px was lower at 1 month and the amount of the PCO increased at 2 months. **Conclusions**: The intake of a HFD or LFD of different durations can influence the establishment of PIR or BIR, and oxidative damage in the fat tissue and cerebral cortex can play an important role.

## 1. Introduction

Insulin resistance refers to the lack of response of insulin receptors to the presence of this hormone, which leads to alterations in the subsequent signaling pathway. As a result, the cellular response to insulin is impaired [1]. Insulin resistance can occur at both the central and peripheral levels [2]. In the case of brain insulin resistance (BIR), it is associated with disruptions in the regulation of thermogenesis, cognitive alterations, and dementia, which are linked to the development of neurodegenerative diseases such as Alzheimer’s disease [3,4,5]. On the other hand, peripheral insulin resistance (PIR) is associated with the development of metabolic alterations such as hyperglycemia and dyslipidemia, which may contribute to the development of type 2 diabetes mellitus [6,7,8]. PIR is a major cause of metabolic diseases commonly associated with central obesity [9].

It has been proposed that in central obesity, white adipose tissue is infiltrated by activated macrophages (M1) [10,11] and correlates with secreting monocyte chemoattractant protein-1 (MCP-1), tumor necrosis factor α (TNFα), interleukin-1β (IL-1β), interleukin 6 (IL-6), and interleukin 1α (IL-1α) [7,12,13]. Additionally, obesity leads to elevated levels of circulating free fatty acids (FFAs), which contribute to reactive oxygen species (ROS) production, exacerbating the adverse effects of metabolic stress associated with obesity [14,15]. The combination of decreased antioxidant defenses and increased ROS levels leads to oxidative stress on macromolecules including lipids, proteins, and nucleic acids [16,17,18,19,20].

The proinflammatory cytokines activate stress-responsive kinases such as c-Jun N-terminal kinase (JNK/SAPK) [21,22]. This activation results in serine phosphorylation surpassing the tyrosine phosphorylation of the insulin receptor (IR), hindering insulin receptor substrate (IRS) binding [22,23,24]. Serine phosphorylation at 307 sites of IRS-1 further impedes phosphoinositide 3-kinase (PI3K) and AKT/PDK activation, ultimately leading to insulin resistance [1,25,26]. Peripherally, in muscle, insulin resistance disrupts glucose entry for energy homeostasis and energy storage [1,26]. In adipose tissue, it hampers glycerol-3-phosphate formation, long-chain fatty acid uptake, triglyceride absorption, lipid esterification, and storage [27].

Recent studies have established the role of insulin in the brain, and it has been shown that insulin enhances N-methyl-D-aspartate (NMDA) and α-amino-3-hydroxy-5-methyl-4-isoxazolepropionic acid (AMPA) receptor activity in rat hippocampal neurons [28,29,30], promotes synaptic plasticity [31], regulates body temperature via the preoptic area (POA) [32], and modulates appetite [33]. Therefore, BIR might affect cognitive function, neurotransmission, thermogenesis, appetite, and body weight, underscoring the need to study these processes. Recently, it has been proposed that in neurodegenerative diseases, such as Alzheimer’s disease, insulin resistance could play a role in the development of dementia [3,5,34].

Numerous studies indicate that a high-fat diet (HFD) induces body weight gain and, subsequently, obesity [8,35,36]; for example, Maciejczyk and collaborators (2018) showed that elevated plasma and cerebral cortex levels of malondialdehyde (MDA)—a marker of lipid oxidative damage—alongside increased glutathione peroxidase (GSH-Px), catalase (CAT), and superoxide dismutase (SOD) activity are established after 8 weeks in animals fed with a HFD [37]. Bodur and collaborators (2019), in mice fed with a HFD for 16 weeks, have shown oxidative damage in lipids subsequent to increased oxidative damage in epididymal adipose tissue [38]. Moreover, Alcalá and collaborators (2017) have demonstrated that HFD intake for 20 weeks induces an increase in ROS together with CAT and SOD activity, resulting in a protective effect on brown adipose tissue [39].

On the other hand, studies using a high-carbohydrate diet (low-fat diet, LFD) in male Wistar rats fed for 3 weeks show increased lipid peroxidation and decreased SOD and glutathione reductase (GSR) activity in the hippocampus [40], and when the LFD intake is extended for 8 weeks, it induces a decrease in antioxidant enzymes, such as GSH-Px and SOD-1, and an increase in prooxidant enzymes such as xanthine oxidase and NADPH oxidase in the hypothalamus of rats fed with fructose [41]. LFD intake for 12 weeks induces hyperglycemia, hyperinsulinemia, and ROS and MDA production in the liver of Sprague Dawley rats [42]; and in C57BL/6 mice fed for 16 weeks, LFD intake reveals decreased MDA levels in subcutaneous white adipose tissue [38].

These findings suggest that insulin resistance could be linked to oxidative damage which, in turn, could target peripheral and central tissues subsequent to HFD or LFD feeding, despite the duration of exposure.

Therefore, this study aimed to determine whether the short-term (1–3 months) consumption of a HFD or LFD induces PIR and/or BIR, and whether theses alterations correlate with an increase in oxidative damage markers in several peripheral tissues and the cerebral cortex of C57BL/6 mice.

## 2. Materials and Methods

### 2.1. Animals and Group Assignment

Male C57BL/6 mice aged 12 weeks (20–25 g weight) were obtained from the vivarium of the Faculty of Chemical Biological Sciences of the Autonomous University of Guerrero. Mice were individually housed under standard conditions of constant room temperature (26 ± 1 °C) and a 12:12 light/dark cycle (lights on at 8 a.m., light onset  =  Zeitgeber time 0, ZT0). They had ad libitum access to food and water.

The mice were randomly assigned to different experimental groups and were fed according to the following categories:(1)Standard diet (C) (LabDiet 5001, St. Louis, MO, USA): providing 29% kcal from protein, 13% kcal from fat, 58% kcal from carbohydrate, and a calorie density of 2.89 kcal/g.(2)Low-fat diet (LFD or high-carbohydrate diet) (Research Diets, Inc., Cat. num. D12450J, New Brunswick, NJ, USA): providing 20% kcal from protein, 10% kcal from fat, 70% kcal from carbohydrates, and a calorie density of 3.82 kcal/g.(3)High-fat diet (HFD) (Research Diets, Inc., Cat. num D12492, New Brunswick, NJ, USA): providing 20% kcal from protein, 60% kcal from fat, 20% kcal from carbohydrates, and a calorie density of 5.21 kcal/g (Appendix A).

The groups were assigned based on the duration of diet exposure as follows:Group 1 (C1, LFD1, HFD1): mice fed for one month.Group 2 (C2, LFD2, HFD2): mice fed for two months.Group 3 (C3, LFD3, HFD3): mice fed for three months.

Body weight was measured weekly (*n* = 5–8). Mice in the HFD group with a body weight difference of >10 g compared to the control group were classified as obese. A summary of the experimental design and the timing of the interventions are shown in Figure 1.

### 2.2. Evaluation of Peripheral Insulin Resistance

To determine the timing of PIR, a glucose tolerance test (GTT) [43] and insulin tolerance test (ITT) [44] were conducted at 2 (C1, LFD1, and HFD1), 6 (C2, LFD2, and HFD2), and 10 (C3, LFD3, and HCD3) weeks of the feeding diet. These tests evaluated insulin sensitivity and the ability to lower fasting blood glucose levels.

Animals were fasted for 6 h prior to blood glucose measurements (at ZT6), which were performed using the OneTouch^®^ UltraTM diagnostic kit (Johnson & Johnson Medical, Saltillo, México, S.A de C.V) according to the manufacturer’s instructions. During the GTT, glucose (2 g/kg body weight, Sigma-Aldrich, St. Louis, MO, USA) was administered orally. For the ITT, animals received an intraperitoneal injection of human recombinant insulin (0.5 IU/kg body weight, Eli Lilly, Humulin R, Mexico). Blood glucose concentrations were monitored at 30, 60, 90, and 120 min post-administration of glucose or insulin.

### 2.3. Evaluation of Brain Insulin Resistance

Telemetry studies were performed to determine BIR and to monitor changes in core body temperature following insulin injections into the POA [32]. For the telemetry studies, peritoneal implant of the calibrated temperature datalogger (SubCue, Calgary, AB, Canada, sensitivity of 0.01 °C) was performed under anesthesia with sodium pentobarbital (SP, 30 g/kg body weight. SP, PiSA Agropecuaria, Hgo, Mex). This surgical procedure, involving laparotomy, was carried out at 2 (C1, LFD1, and HFD1), 5 (C2, LFD2, and HFD2), and 9 (C3, LFD3, and HFD3) weeks after diet initiation. The dataloggers were programmed using SubCue Analyzer software (Alberta, Canada) to record the core body temperature every 10 min for 15 days. The information for this miniature temperature data logger and hardware was obtained from SubCue (http://www.subcue.com/). Following surgery, the mice were individually housed in standard Plexiglas boxes.

For the administration of insulin directly into the POA, mice underwent stereotaxic surgery [32]. Anesthesia was induced with SP, and the mice were secured in a stereotaxic frame (Model 902 Dual Small Animal Stereotaxic Instrument, David Kopf Instruments, Tujunga, CA, USA). A guide cannula (26 G, 10 mm length) was implanted into the mouse skull at coordinates anterior/posterior from bregma +0.34 mm, lateral = midline, and ventral = 4.4 mm. This surgical procedure was carried out at 3 (C1, LFD1, and HFD1), 7 (C2, LFD2, and HFD2), and 11 (C3, LFD3, and HFD3) weeks after diet initiation. Two days after surgery, insulin injections were performed into the POA using a microsyringe (5 μL, Hamilton Company, Reno, NV, USA) connected to the implanted cannula and plastic tubing. Recombinant human insulin (0.015, 0.03, and 0.06 IU) or vehicle [artificial cerebrospinal fluid (ACSF), prepared according to Alzet^®^ guidelines] were injected randomly on different days starting at 2 p.m. (ZT6). The injections were administered over 1 min to allow for hormone diffusion. At the end of the time periods, the device was recovered in the sacrifice. The core body temperature data were extracted using SubCue Analyzer software for analysis.

### 2.4. Sacrifice and Tissue Sampling

The animals were euthanized at the end of each experimental period (4, 8, and 12 weeks) to collect blood and tissues for the assessments of oxidative damage. Before sacrifice, the mice were anesthetized with SP. Blood was collected via cardiac puncture (approximately 700 to 1000 μL), transferred to microtubes containing EDTA, and then centrifuged at 3500 rpm for 10 min at 4 °C to obtain plasma. Tissues were dissected anatomically, comprising the cerebral cortex and approximately 30 mg each of subscapular brown adipose tissue and visceral white adipose tissue. These tissues were placed in individual tubes containing 450 μL of cold PBS. Subsequently, the tissues were homogenized. The homogenates were then centrifuged at 3500 rpm for 10 min at 4 °C to obtain the supernatant for the analysis of oxidative damage markers.

### 2.5. Glutathione Peroxidase (GSH-Px) Activity

For the determination of GSH-Px activity [45], the samples were incubated in a redox solution (50 mM PBS, pH 7.0, 1 mM EDTA, 1 mM sodium azide, 0.2 mM NADPH, 1 U/mL glutathione reductase, and 1 mM reduced glutathione) and incubated at 37 °C for 5 min. The reaction was started by adding 0.25 mM H_2_O_2_. The absorbance reading was carried out at a wavelength of 340 nm for 3 min in 1 min intervals in the NanoDrop at 37 °C. The millimolar absorption coefficient of NADPH at 340 nm (6.22 × 10^6^) was used to quantify the enzymatic activity of GSH-Px.

### 2.6. Malondialdehyde (MDA) Concentration

MDA was used to quantify lipid peroxidation. Colorimetric determinations of MDA were performed with 1-methyl-2-phenylindole (MPI, Sigma-Aldrich) in a HCl-based assay, and values were calculated from a standard curve using 1,1,3,3 tetramethoxypropane (Sigma-Aldrich) [46].

Determination of the MDA levels was performed by adding 50 μL of sample, 50 μL of H_2_O, 325 μL of 10 mM MPI, and 100 μL of 37% hydrochloric acid in dark microtubes. The mixture was incubated at 45 °C for 40 min. The MPI for the control tube was replaced with HCl at 20 mM. Subsequently, samples were centrifuged at 7000 rpm for 10 min, the supernatant was obtained, and the absorbance was read at 586 nm. Data are reported in μM and were adjusted per mg of protein.

### 2.7. Determination of the Protein Carbonyl Group (PCO)

The content of the PCO was used to evaluate the damage to proteins in plasma and cerebral cortex through the derivatization of 2,4-dinitrophenylhydrazine (DNPH, Sigma-Aldrich) [47]. A total of 50 μL of sample and 10 mM DNPH were placed in glass test tubes (1.5 µL), and the mixture was incubated in the dark at room temperature for 1 h. Subsequently, the proteins were precipitated with 5% and 2.5% trichloroacetic acid (*w*/*v*) and centrifuged at 3500 rpm for 10 min. The pellets were washed with ethanol/ethyl acetate (1:1; *v*/*v*) and were resuspended in 500 μL of guanine hydrochloride (6 M) at 37 °C for 10 min. The level of the PCO was quantified with spectrophotometry at 370 nm using an extinction coefficient of 22,000 M^−1^cm^−1^. The results are reported as nM of osazones and adjusted per mg of protein.

The PCO in adipose tissue was determined with the simplified alkaline DNPH derivatization methodology [48], incorporating the modifications of Lo and collaborators [49]. A total of 100 μL of sample and 100 μL of 10 mM DNPH were placed in glass test tubes, and the mixture was incubated for 10 min at room temperature. Subsequently, 50 μL of 1 M NaOH was added and incubated for 10 min at room temperature. Finally, absorbance readings were taken at 450 nm. The results were obtained through the molar extinction coefficient of DNPH under alkaline conditions of 22.308 M^−1^cm^−1^ and are reported as nM of osazones and adjusted per mg of proteins.

### 2.8. Statistical Analysis

The data were expressed as the mean and standard deviation. Variations between groups were evaluated with two-way analysis of variance (ANOVA) for a global comparison of the groups, followed by a Student’s *t*-test to analyze differences between individual experimental groups (C vs. LFD and C vs. HFD). The area under the curve (AUC) and statistical analysis were performed using the GraphPad Prism (v 8) software. A value of *p* < 0.05 was considered statistically significant.

## 3. Results

### 3.1. Obesity Induction by HFD

A HFD and LFD were used to create models of obesity and insulin resistance in C57BL/6 mice. Due to laparotomy and stereotaxic surgery performed at different times (indicated by arrows in the graphs for each group), the mice did not gain weight immediately following the laparotomy.

After 1 month of consumption of the diets, no significant changes in body weight were observed (Figure 2A) among the groups (HFD1, LFD1, and C1). In the animals fed with the diets for 2 months, it was observed that the HFD2 mice experienced a significant weight increase of 7.6 g (32.4 g ± 2.0 g, *p* = 0.0037) compared to the C2 mice (24.8 g ± 0.5 g) (Figure 2B). Finally, when the animals consumed the diets for 3 months, we observed that the HFD3 mice showed a significant increase in body weight of 12.7 g (38.6 g ± 2.3 g, *p* = 0.0004) compared to the C3 mice (25.9 g ± 1.1 g) (Figure 2C). The growth rate was clearer at the end of the experimental period at 2 and 3 months.

### 3.2. HFD Causes Glucose Intolerance in C57BL/6 Mice Starting from the First Month of This Diet

The GTT was conducted and the AUC was calculated to assess changes in glucose tolerance in mice as a potential factor for developing PIR. Blood glucose levels were measured at 0, 30, 60, 90, and 120 min after oral glucose administration, in groups of animals that had been fed different diets for 1 month (Figure 3A), 2 months (Figure 3B), and 3 months (Figure 3C). The AUC of the GTT is shown in Figure 3D.

In the first month of diet exposure, the fasting glucose level (198 ± 48.0 mg/dL) and the AUC (AUC = 27, 636 ± 2314, *p* < 0.001) of the HFD1 group were higher than those of the C1 group (140.6 ± 20.7 mg/dL; AUC = 21.309 ± 1.599) (Figure 3A–D). In the second month of diet exposure, the fasting glucose level of the HFD2 mice (187.8 ± 45.3 mg/dL) and the AUC (AUC = 26,415 ± 1958, *p* < 0.001), as well as the fasting glucose level (188.5 ± 34.9 mg/dL) and AUC (AUC= 24,925 ± 1880, *p* = 0.0025) of the LFD2 mice, were higher compared to the C2 group (147.8 ± 27.6 mg/dL) (Figure 3B–D). In the third month, the fasting glucose level (194.0 ± 31.3 mg/dL) and the AUC (AUC = 25,654 ± 1944, *p* < 0.001) of the HFD3 mice were higher than those of the C3 mice (140.0 ± 24.4 mg/dL, AUC= 21,448 ± 1156) (Figure 3C,D).

Thirty minutes after glucose administration, the glucose levels increased to 279.2 ± 69.7 mg/dL, 259.2 ± 54.6 mg/dL, and 289.4 ± 68.4 mg/dL for HFD1, HFD2, and HFD3, respectively. Although the levels decreased over the next 60 min, they remained above approximately 200 mg/dL. The data indicated the presence of glucose intolerance in these groups compared to the C group (Figure 3A–C).

Regarding the GTT result for LFD mice, it was higher only in the LFD2 group (AUC = 24.925 ± 1.880, *p* = 0.0025) compared to the C2 control group (AUC = 21.355 ± 1.592) (Figure 3D).

### 3.3. Peripheral Insulin Resistance Is Established in C57BL/6 Mice Fed HFD up to the Third Month of Administration

To determine if the animals develop insulin insensitivity as an indicator of PIR, blood glucose levels were evaluated after insulin administration (0.5 IU/kg) for 2 h with subsequent glucose measurements at 0, 30, 60, 90, and 120 min in groups of animals that had been fed different diets for 1 month (Figure 3E), 2 months (Figure 3F), and 3 months (Figure 3G). The AUC of the ITT is shown in Figure 3H.

Animals fed the HFD responded to the insulin during the first (HFD1, T^0 min^ = 205.6 ± 45.7 mg/dL, T^30 min^ = 149.6 ± 52.5 mg/dL) and second (HFD2, T^0 min^ = 182.8 ± 35.5 mg/dL, T^30 min^ = 82.6 ± 43.2 mg/dL) months by decreasing glucose levels after insulin injection, as compared to the respective controls (C1, T^0 min^ = 155.6 ± 46.1 mg/dL, T^30 min^ = 99.6 ± 36.1 mg/dL; C2, T^0 min^ = 149.0 ± 19.6 mg/dL, T^30 min^ = 87.0 ± 28.6) (Figure 3E,F).

However, in the HFD3 group, the animals did not show a decrease in glucose levels after insulin injection by maintaining linear levels of 180 mg/dL during the 2 h of the test, and the AUC of the ITT was higher (AUC = 19,614 ± 1365, *p* < 0.001) compared to C3 (T^0 min^ = 134.3 ± 23.3 mg/dL, T^30 min^ = 100.3 ± 24.7 mg/dL, AUC = 13,670 ± 808) (Figure 3G,H), confirming insulin insensitivity in this experimental group.

With respect to the LFD mice, they showed similar changes in glucose levels under the influence of insulin with their respective controls (Figure 3E,F). Despite this, the AUC of the ITT was higher in the LFD1 (AUC = 15,110 ± 1427, *p* < 0.001) and LFD3 (AUC = 17,014 ± 944, *p* < 0.001) mice compared to C1 (AUC = 12,438 ± 1711) and C3 (AUC= 13,670 ± 808) (Figure 3H).

### 3.4. LFD Diet Promotes BIR in the First Two Months of Exposure

To determine the presence of BIR, insulin (0.015, 0.03, and 0.06 IU) was administered into the POA after stereotaxic surgery, and the change in body temperature was detected using the datalogger device previously implanted using abdominal surgery. Temperature was detected for 4 h after insulin administration in the experimental groups. In the mice administrated with ACFS and fed with C, HFD, and LFD, there were no significant changes (Figure 4A,E,I).

In the first month (Figure 4B–D), the temperature at time zero of the C1 mice was 35.82 ± 0.37 °C; for LFD1, it was 36.02 ± 0.17 °C; and for HFD1, it was 36.45 ± 0.13 °C. At 30 min after the injection of 0.015 IU of insulin in the POA, the temperature was higher in the HFD1 mice (36.95 ± 0.01, *p =* 0.0108) compared to the C1 mice (36.52 ± 0.13 °C), while in the LFD1 mice, the temperature was 36.57 ± 0.25 °C (*p* = 0.9798). However, at 170 min after administration, the temperature of the LFD1 mice (35.57 ± 0.2 °C, *p* = 0.0424) was lower compared to the C1 mice (36.46 ± 0.3 °C) and the HFD1 group (36.37 ± 0.19 °C).

Interestingly, the hyperthermic effects of insulin were also not observed in the LFD2 animals (Figure 4E–H), which maintained a temperature of 36.48 ± 0.2 °C (30 min) and 36.43 ± 0.3 °C (120 min) after the 0.015 IU insulin injection. When comparing the temperature values at 170 min with the C2 (36.88 ± 0.3 °C) and HFD2 (36.6 ± 0.2 °C) groups, we observed that the LFD2 mice (36.05 ± 0.2 °C) presented a lower temperature (*p* = 0.0208) (Figure 4F).

At 3 months, there were no significant differences between the C3, LFD3, and HFD3 groups (Figure 4I–L). These observations suggest that the alteration in the thermogenic capacity of insulin observed in the LFD1 and LFD2 groups was reversed.

Based on the real temperature values, it is demonstrated that the LFD did not maintain the increase in temperature in the mice when 0.015 IU of insulin was administered, which may indicate early signs of insulin resistance in the brain.

### 3.5. Oxidative Damage of Lipids and Proteins, and Enzymatic Activity in Plasma

To evaluate oxidative damage in the bloodstream and different tissues, we used samples of plasma, white adipose tissue (visceral), brown adipose tissue, and the cerebral cortex. As markers of oxidative damage, we evaluated MDA as a marker of lipid peroxidation, the PCO as a product of protein oxidation, and GSH-Px activity as an indicator of the antioxidant system. Our results are expressed as a percentage (%), considering the mice fed a standard diet as the reference group (control group, C). In the plasma, the MDA and PCO levels, and GSH-Px activity showed no significant changes in any of the study groups (Figure 5A–C).

### 3.6. Oxidative Damage of Lipids and Proteins, and Enzymatic Activity in White Adipose Tissue

In white adipose tissue, the level of MDA was variable according to the type of diet and duration of consumption. After 1 month of feeding with different diets, MDA levels showed no significant changes in the LFD1 and HFD1 mice compared to the C1 mice. However, after 2 months of following different diets, MDA levels increased by 104 ± 47.3% (*p* = 0.0434) in the LFD2 mice and 437 ± 94.9% (*p* = 0.0056) in the HFD2 mice compared with the C2 mice (106.2 ± 10.5%). Conversely, after 3 months of following different diets, MDA levels decreased by 75 ± 15.4% (*p* = 0.0375) in the LFD3 mice compared to the C3 mice (106.7 ± 27.3%) (Figure 5D).

Regarding PCO levels and GSH-Px, they showed no significant changes in any of the study groups (Figure 5E,F).

### 3.7. Oxidative Damage of Lipids and Proteins, and Enzymatic Activity in Brown Adipose Tissue

In brown adipose tissue, the level of MDA significantly decreased by 52 ± 8.6% (*p* = 0.0006) in the LFD1 mice and 50 ± 7.8% (*p* = 0.0431) in the LFD2 mice compared to their respective control groups (C1, 100.0 ± 5.9%; C2, 102.6 ± 23.5%). Likewise, the MDA level in the HFD mice showed no significant changes (Figure 5G).

The PCO level and the enzymatic activity of GSH-Px showed no significant changes in any of the groups (Figure 5H,I).

### 3.8. Oxidative Damage of Lipids and Proteins, and Enzymatic Activity in the Cerebral Cortex

In the cerebral cortex, the level of MDA was significantly decreased by 36 ± 9.5% (*p =* 0.0275) in the HFD1 mice compared to the C1 mice (100 ± 10.1%). However, in the groups fed for 2 and 3 months with different diets, there were no changes in the level of MDA (Figure 5J). Additionally, the PCO level only increased 102 ± 35.9% (*p =* 0.0383) in the LFD2 mice compared to the C2 mice (103.2 ± 10.5%) (Figure 5K).

Finally, GSH-Px activity decreased by 79.2 ± 5.9% (*p =* 0.0329) in the LFD1 mice compared to the C1 mice (100 ± 8.6%) (Figure 5L).

## 4. Discussion

In the present study, we investigated the effects of a HFD, LFD, or standard diet on the establishment of PIR or BIR using a murine model of diet-induced obesity in C57BL/6 mice. We also examined the presence of diet-induced oxidative damage markers in the establishment of BIR, PIR, or both.

HFD-fed mice reached significant weight gain at 3 months. As previously reported, animals fed with a HFD gained weight after a week of diet initiation compared to the controls, confirming the role of a HFD in the development of obesity [8,35,36,50,51]. Furthermore, the GTT indicated the development of glucose intolerance in animals fed a HFD for 1, 2, or 3 months, with consistently elevated glucose levels compared to the controls (C mice). Specifically, the HFD3 animals exhibited sustained hyperglycemia compared to the LFD3 and C3 groups, indicating disrupted glucose homeostasis, which correlates with their gradual weight gain. Additionally, glucose levels in the HFD3 group remained elevated after insulin injection during the ITT, confirming the establishment of PIR.

PIR after 3 months of HFD consumption may be attributed to the increased availability of FFA from hypertrophied adipocytes [13] and the elevated concentration of DAG and sphingolipids, such as ceramide. DAG is a molecule that impairs insulin signaling through the activation of several isoforms of protein kinase C (PKC), leading to PIR through increased IRS-1 serine^1101^ phosphorylation and the inhibition of insulin-stimulated IRS-1 tyrosine phosphorylation and Akt2 phosphorylation. Ceramide, on the other hand, interferes with the insulin signaling pathway by inhibiting the insulin-induced phosphorylation of Akt/PKB and the activation of protein phosphatase 2A, resulting in impaired insulin sensitivity in skeletal muscle and liver. Ceramide also causes endoplasmic reticulum stress and mitochondrial dysfunction, which increase the concentration of ROS and, thus, the development of oxidative stress [13,52].

ROS induce the phosphorylation of IR and IRS on serine residues which attenuate insulin signaling, hindering the translocation of GLUT4 to the cell membrane, thus impairing glucose uptake, which leads to an increase in blood glucose and insulin resistance [33,53]. Additionally, released FFA and oxidative stress from hypertrophic adipose tissue inhibit the antilipolytic action of insulin [54,55]. Therefore, HFD intake is related to glucose intolerance and PIR.

In this study, plasma, white adipose tissue from the visceral region, and brown adipose tissue from the scapular area were evaluated to determine the presence of oxidative damage at the peripheral level.

Although we did not find significant differences in plasma, we did observe them in white adipose tissue. The HFD2 and LFD2 mice showed higher levels of MDA in visceral white adipose tissue compared to the C2 mice, with no changes in the activity of GSH-Px, an enzyme belonging to the endogenous antioxidant system responsible for catalyzing the replacement of glutathione disulfide using H_2_O_2_ [56]. Our results suggest that lipid damage is present without an increase in GSH-Px activity, which could counteract the damage, indicating the first signs of oxidative stress in visceral white adipose tissue.

This behavior was also observed by Alcalá and collaborators (2017), who found an increase in the level of ROS without significant differences in GSH-Px activity in the visceral white adipose tissue of mice fed with a HFD for 5 months [39].

Regarding oxidative damage in brown adipose tissue, the results show decreased MDA levels in mice fed LFD1 and LFD2 compared to control mice, with no changes in GSH-Px. This suggests that oxidative damage might not occur, because this tissue has adipocytes with a greater number of mitochondria compared to white adipose tissue [57]. In the mitochondria of brown adipocytes, the uncoupling protein-1 (UCP1) is expressed, which is a membrane protein that carries out the exchange of electrons for heat generation and reduces oxidative stress generation in this tissue. Additionally, brown adipose tissue may exhibit resistance to oxidative stress and damage even under HFD and LFD diets due to the activity of the enzymes of the antioxidant system [58] as CAT and SOD [39].

Conversely, in this study, LFD intake was associated with BIR, as mice fed with an LFD for 1 and 2 months did not exhibit hyperthermia in response to 0.015 IU of insulin. Insulin causes POA neurons to signal to the dorsomedial nucleus and, in turn, to the raphe pallidus, which stimulates the sympathetic nervous system to release norepinephrine from postganglionic neurons, which acts on brown adipose tissue and triggers UCP1. This is then activated to release thermal energy by taking advantage of the proton gradient in the inner mitochondrial membrane and increasing core body temperature [59,60]. Thus, BIR was evidenced by the inability of insulin to induce thermogenesis in LFD-fed animals [32]. In these animals, the oxidative damage at the brain level was evidenced by the increased PCO (LFD2 mice) and decreased GSH-Px (LFD1 mice).

We consider that in these mice, the high percentage of carbohydrates (70%) of the LFD promote hyperglycemia, which, in turn, stimulates insulin secretion by the β cells of the pancreas, leading to hyperinsulinemia [6], which causes large amounts of insulin to be transported to the brain, causing the desensitization of IR [61]. In addition to hyperinsulinemia, hyperglycemia is also present in this cellular environment. In POA neurons and astrocytes, the glucose transporters GLUT3 and GLUT1 are expressed. These transporters do not require insulin stimulation to facilitate glucose entry into the cell [62,63,64], resulting in higher intracellular glucose concentration, increased glycolysis, and the elevated production of ROS during mitochondrial respiration, ultimately triggering oxidative damage [65,66]. However, we did not observe any alteration that would indicate the presence of oxidative damage in LFD3. Coincidentally, these animals did not exhibit BIR as they again responded to the insulin stimulus in the POA by showing hyperthermia. This may be due to a state of neuronal preconditioning. Some studies showed that an excessive intake of carbohydrates or fat leads to increased ROS production, which triggers an adaptive response that promotes the production and action of antioxidant enzymes as part of a long-term adaptation process, inducing protective mechanisms. This phenomenon is known as hormesis, whereby cells adapt to adverse conditions [56,67,68]. A moderate ROS concentration promotes neuronal preconditioning, as blocking the production of free radicals increases cell death [67] in such a way that the neuronal function, just as the sensitivity of the IR, could be restored.

The development of BIR is an understudied topic compared to PIR; hence, the timeline of the first evidence of BIR is not precisely known. In this study, we focused on the effect of acute dietary exposure and the appearance of BIR or PIR, observing that from the first month, there are already signs of BIR in LFD-fed mice. While PIR has been studied more extensively, our research shows signs of metabolic alterations from the first month that continue for 3 months in HFD-fed mice. It is worth mentioning that prolonging the exposure time to a HFD can intensify the PIR, as evidenced in other research based on 20 [69,70], 22 [71], and 36 weeks of exposure to a HFD [70]. Interestingly, it has been reported that mice exposed for 52 weeks to a HFD or LFD did not exhibit PIR; there was a regression of this metabolic abnormality [72].

In conclusion, our research allowed us to detect the first indications of BIR, PIR, or both in relation to the intake of different diets, as well as the association between diet consumption and the presence of markers of oxidative stress.

## Figures and Tables

**Figure 1 biomedicines-13-00404-f001:**
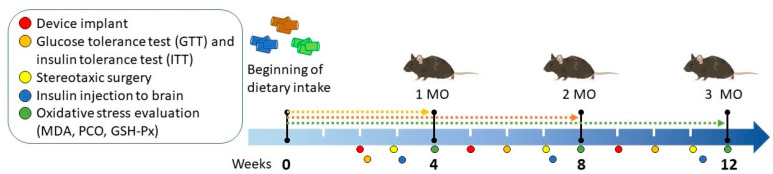
Experimental diagram of obesity induction and measurement of peripheral and cerebral insulin resistance in C57BL/6 mice. Mice were exposed to standard, low-fat, or high-fat diets for 1 (4 weeks), 2 (8 weeks), and 3 (12 weeks) months (MO). Body weight was measured weekly until 12 weeks. Device implant was performed at weeks 2 (1 MO group), 5 (2 MO group), and 9 (3 MO group). Subsequently, the insulin tolerance test (ITT) and the glucose tolerance test (GTT) were performed at weeks 2, 6, and 10. Placement of the guide cannula for insulin injections in the mice was performed through stereotaxic surgery at weeks 3, 7, and 11. Finally, the mice were sacrificed at weeks 4, 8, and 12 to obtain organs and determine oxidative damage markers.

**Figure 2 biomedicines-13-00404-f002:**
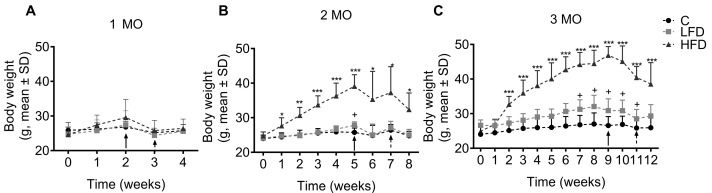
Effect of fat intake on body weight of C57BL/6 mice. Body weight of mice fed different diets for 1 (**A**), 2 (**B**), and 3 (**C**) months (MO) are shown. Data are shown as the mean ± standard deviation. Statistical significance using two-way analysis of variance (ANOVA) followed by Student’s *t*-test. C: standard diet (black circle); LFD: low-fat diet or high-carbohydrate diet (light-gray square); HFD: high-fat diet (strong gray triangle). The solid arrow indicates the time of performance of the laparotomy and the dotted arrow indicates the time of performance of stereotaxic surgery. C vs. LFD = + *p* < 0.05, C vs. HFD ** p* < 0.05, ** *p* < 0.01, and *** *p* < 0.001.

**Figure 3 biomedicines-13-00404-f003:**
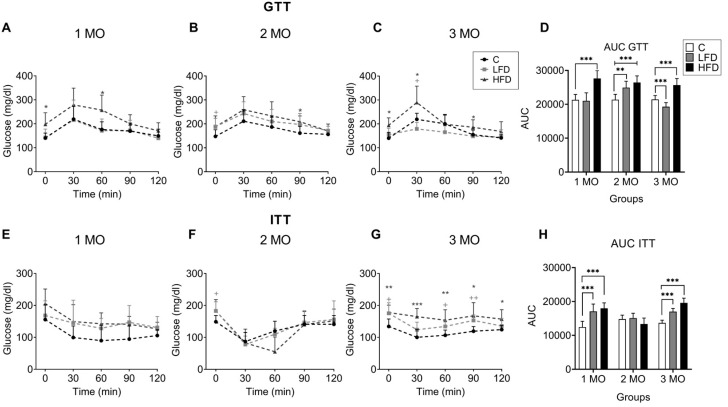
Effect of HFD and LFD intake on glucose tolerance and peripheral insulin sensitivity in C57BL/6 mice. Glucose levels are shown versus time (0, 30, 60, 60, 90, and 120 min) after administration of 2 g glucose/kg body weight in mice fed for 2 (A), 6 (B), and 10 (C) weeks on low-fat or high-carbohydrate diet (LFD), high-fat (HFD), or standard diet (C) and the area under the curve (AUC) (D) for the glucose tolerance test (GTT) for the 1, 2, and 3 month (MO) groups. Glucose levels are shown versus time (0, 30, 60, 60, 90, and 120 min) after administration of 0.5 IU insulin/kg body weight intraperitoneally in LFD, HFD, or C mice fed for 2 (E), 6 (F), and 10 (G) weeks and the AUC (H) for the insulin tolerance test (ITT) for the 1, 2, and 3 MO groups. Data are shown as the mean ± standard deviation. + *p* < 0.05, ++ *p* < 0.01 significance LFD vs. C. * *p* < 0.05, ** *p* < 0.01, *** *p* < 0.001 significance HFD vs. C.

**Figure 4 biomedicines-13-00404-f004:**
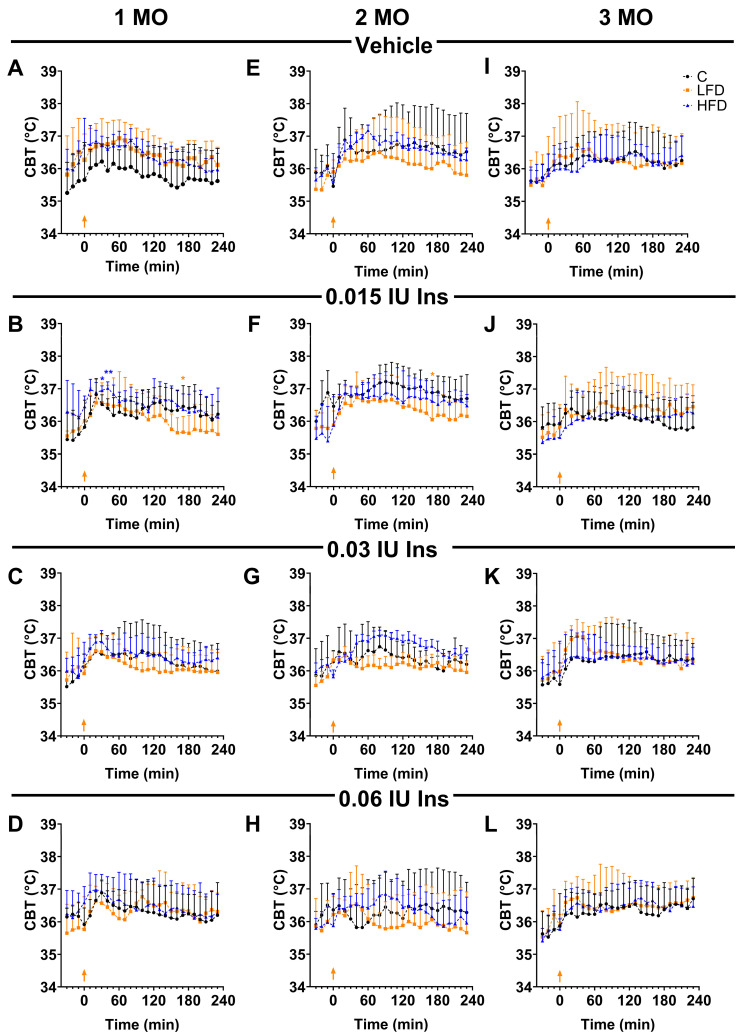
Effect of brain insulin injections on core body temperature in C57BL/6 mice. Changes in core body temperature (CBT) over 4 h after local injection of insulin (0.015, 0.03, and 0.06 IU) and vehicle in the POA by groups of 1 (**A**–**D**), 2 (**E**–**H**), and 3 (**I**–**L**)months (MO) of exposure to the diets. Data are shown as the mean ± standard deviation. * *p* < 0.05, ** *p* < 0.01 significance vs. C. C: standard diet (black circle); LFD: low-fat diet or high-carbohydrate diet (orange square); HFD: high-fat diet (blue triangle). The orange arrow indicates insulin injection time.

**Figure 5 biomedicines-13-00404-f005:**
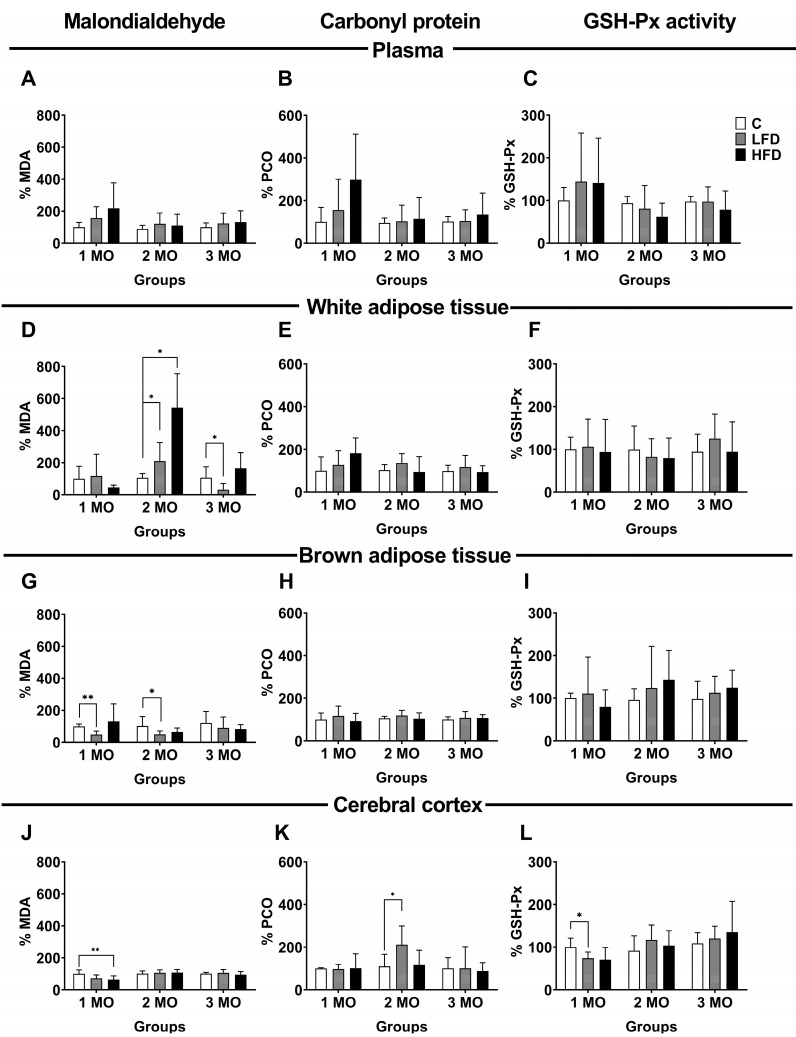
Oxidative damage in plasma, white adipose tissue, brown adipose tissue, and cerebral cortex. Percentage of malondialdehyde (MDA) (**A**,**D**,**G**,**J**), protein carbonyl group (PCO) (**B**,**E**,**H**,**K**), and glutathione peroxidase activity (GSH-Px) (**C**,**F**,**I**,**L**) in plasma, white adipose tissue, brown adipose tissue, and cerebral cortex of mice fed a standard diet (C), low-fat diet (LFD), and high-fat diet (HFD) for 1, 2, and 3 months (MO). Data are shown as the mean ± standard deviation. Statistical significance with two-way analysis of variance (ANOVA) followed by Student’s *t*-test. * *p* < 0.05; ** *p* < 0.01.

## Data Availability

The raw data supporting the conclusions of this article will be made available by the authors on request.

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
