# Peer review of "Relationship Between Brain Insulin Resistance, Carbohydrate Consumption, and Protein Carbonyls, and the Link Between Peripheral Insulin Resistance, Fat Consumption, and Malondialdehyde"

_biomedicines, 2025, doi:10.3390/biomedicines13020404_

Round 1

Reviewer 1 Report

Comments and Suggestions for Authors

The manuscript entitled "Brain insulin resistance is related to carbohydrate consumption and the presence of carbonylated proteins, while peripheral insulin resistance is linked to fat consumption and the presence of malondialdehyde" is a well designed study, which is strongly supported by the hypothesis, methodology and writing. I suggest some modifications prior to acceptance. My comments are as follows

1. Authors should describe both peripheral and brain insulin resistance and their differences briefly in first paragraph of introduction.

2. In analysis, since authors are focusing on insulin resistance, I recommend to analyze HOMA IR index which is the best indicator of insulin resistance. 

3. Authors could have done the analysis of genes associated with insulin resistance in the tissues.

4. Authors shall include the details of model and make of softwares, place and country.

Author Response

The manuscript entitled "Brain insulin resistance is related to carbohydrate consumption and the presence of carbonylated proteins, while peripheral insulin resistance is linked to fat consumption and the presence of malondialdehyde" is a well designed study, which is strongly supported by the hypothesis, methodology and writing. I suggest some modifications prior to acceptance. My comments are as follows

  1. Authors should describe both peripheral and brain insulin resistance and their differences briefly in first paragraph of introduction.

A: We included the requested information in the first paragraph.

  1. In analysis, since authors are focusing on insulin resistance, I recommend to analyze HOMA IR index which is the best indicator of insulin resistance. 

A: Thank you for your comment. We acknowledge that HOMA-IR is widely recognized as an indicator of insulin resistance, however it requires fasting glucose and insulin levels. Unfortunately, in our study, the absence of fasting insulin levels limits our ability to calculate HOMA-IR directly. On the other hand, the HOMA-IR model evaluates fasting glucose/insulin levels, and is an indicator of the interplay between liver and pancreas function. Although it is a useful method to determine the metabolic state of the individual, it is limited by the fact that it does not reflect changes in peripheral insulin sensitivity [1,2].

Additionally, the glucose (GTT) and insulin (ITT) tolerance test are dynamic test. The GTT has the advantage of measuring the decrease in blood glucose, indicating that the cells (influenced by the activity of the muscle and liver) have some capacity to internalize it. Likewise, the ITT evaluates the action of the exogenous hormone in the living model [3–5].

  1. Authors could have done the analysis of genes associated with insulin resistance in the tissues.

A: We appreciate your suggestion. In this work, we aim to provide evidence on the timeline of PIR and BIR development. However, we have also assessed other molecular aspects in the tissues, such as the insulin receptor expression and its tyrosine phosphorylation, to evaluate molecular changes in insulin receptor signaling (Preliminary data not shown in this work, but we have included them as supplementary material).

  1. Authors shall include the details of model and make of softwares, place and country.

A: The C57BL/6 strain is widely used for studies in vivo of obesity, diabetes and metabolic disorders. In addition, there is evidence that male C57BL/6J mice are susceptible to diet-induced obesity, also alterations in glucose homeostasis are more efficient in C57BL/6 males than in females [6–9]. We have added some citations to the methodology and completed the software information as requested.

References

  1. Buchanan, T.A.; Watanabe, R.M.; Xiang, A.H. Limitations in Surrogate Measures of Insulin Resistance. The Journal of Clinical Endocrinology & Metabolism 2010, 95, 4874–4876, doi:10.1210/jc.2010-2167.
  2. Antunes, L.C.; Elkfury, J.L.; Jornada, M.N.; Foletto, K.C.; Bertoluci, M.C. Validation of HOMA-IR in a Model of Insulin-Resistance Induced by a High-Fat Diet in Wistar Rats. Arch. Endocrinol. Metab. 2016, 60, 138–142, doi:10.1590/2359-3997000000169.
  3. Manwani, T.; Wanjari, A.; Acharya, S. Insulin Resistance and Its Detection. Journal of Datta Meghe Institute of Medical Sciences University 2024, 19, 416–420, doi:10.4103/jdmimsu.jdmimsu_171_23.
  4. Pedro, P.F.; Tsakmaki, A.; Bewick, G.A. The Glucose Tolerance Test in Mice. In Animal Models of Diabetes; King, A.J.F., Ed.; Methods in Molecular Biology; Springer US: New York, NY, 2020; Vol. 2128, pp. 207–216 ISBN 978-1-07-160384-0.
  5. Cózar-Castellano, I.; Perdomo, G. Assessment of Insulin Tolerance In Vivo in Mice. In Animal Models of Diabetes; King, A.J.F., Ed.; Methods in Molecular Biology; Springer US: New York, NY, 2020; Vol. 2128, pp. 217–224 ISBN 978-1-07-160384-0.

Reviewer 2 Report

Comments and Suggestions for Authors

Brain insulin resistance is related to carbohydrate consumption and the presence of carbonylated proteins, while peripheral insulin resistance is linked to fat consumption and the presence of malondialdehyde

The study aims to demonstrate whether HFD or LFD induces peripheral insulin resistance (PIR) before brain insulin resistance (BIR), and whether the timing of these alterations correlates with heightened oxidative damage markers in plasma, adipose tissue and cerebral cortex. However, the approach and the overall design of the study are weak. The authors should address the following concerns.

·         The authors should include references to the methods used for all the experimental assays.

·         Why did the authors use both two-way analysis of variance (ANOVA) and followed by a student t-test for statistical comparison? Since there are three experimental groups with two variables, the preferred way of statistical comparison is the two-way analysis of variance.

·         In the 3-month groups, LFD3 showed decreased AUC of GTT even lower than control. What could be the possible reason?

·         What is the possible reason for the normalized ITT only in 2-months for all groups?

·         Line: 455-456-“In the present work, oxidative damage at the brain level was evidenced by PCO increased (LFD2 mice) and decreased GSH-Px (LFD1 mice)”. What happens to the brain in 3 month group? What will be the possible reason for the normalized values of MDA, PCO and GSH-Px after 3 months of LFD?

·         The discussion is too lengthy and describes too many aspects that are not even studied in the present study. The discussion part is more about the findings from other studies, rather than discussing the findings of the present by including previous studies. 

·         Conclusion says that: “The use of HFD or LFD at different duration can influence the establishment of PIR or BIR, and oxidative damage in the fat tissue and cerebral cortex can play an important role”. However, the results of the study do not support the conclusion statement. FOR BIR, only 2 months showed an increase in %PCO in the cerebral cortex of LFD. Also, % of MDA was decreased in the cerebral cortex of LFD after 1 month and no significant differences were observed after subsequent months. With these results, the authors cannot conclude that oxidative stress plays an important role in LFD in BIR. More detailed studies are required to validate the aim.

·         Writing is incomprehensible with plenty of grammatical, typographical and punctuation errors.

·         Overall, the study lack clarity in concluding the aim and looks scientifically weak.

Comments on the Quality of English Language

The English could be improved.

Author Response

Brain insulin resistance is related to carbohydrate consumption and the presence of carbonylated proteins, while peripheral insulin resistance is linked to fat consumption and the presence of malondialdehyde

The study aims to demonstrate whether HFD or LFD induces peripheral insulin resistance (PIR) before brain insulin resistance (BIR), and whether the timing of these alterations correlates with heightened oxidative damage markers in plasma, adipose tissue and cerebral cortex. However, the approach and the overall design of the study are weak. The authors should address the following concerns.

  1. The authors should include references to the methods used for all the experimental assays.

A: We have added citations to the methodology and completed the information as requested.

  1. Why did the authors use both two-way analysis of variance (ANOVA) and followed by a student t-test for statistical comparison? Since there are three experimental groups with two variables, the preferred way of statistical comparison is the two-way analysis of variance.

A: The ANOVA test is used to perform a general analysis of variances, but for a more detailed study, T-student was applied, because this test has a high detectable significant difference. In addition, this test is more powerful when the groups analyzed are small, in this case the n is 5-8 [1]. Considering the above, we are only interested in comparing the means of two groups to determine if there is any statistical difference in those specific comparisons. This also reduces variability when performing the statistical analysis with ANOVA, where more variables are taken into account for analysis. The analysis of the results was performed between C vs LFD and C vs HFD in all experimental groups (1, 2 and 3 months).

  1. In the 3-month groups, LFD3 showed decreased AUC of GTT even lower than control. What could be the possible reason?

A: During GTT, (figure 3A, B and C), glucose levels increase in control mice after carbohydrate administration, whereas in LFD3 mice, there is no change in blood glucose over time. In fact, in LFD3 the levels do not increase. The above, may be due to their constant carbohydrate intake, which leads to  conditioned insulin production by the pancreas and that the response to insulin in tissues such as muscle and liver, allow the efficient regulation of blood glucose homeostasis [2,3]. That is, the organism adapted to consuming large amounts of carbohydrates, a situation that did not occur with animals fed high-fat diets. It is worth noting that, when comparing the ITT plot of LFD3 mice, glucose values do change, demonstrating that the mice respond to exogenous insulin administration [4].

On the other hand, the AUC (Figure 3D) in a glucose tolerance test provides information on how the body handles an oral glucose load [2]. When evaluating the AUC, we observe a decrease in LFD3 mice, which may indicate cell preconditioning in these animals. However, this situation does not occur in all organisms.

  1. What is the possible reason for the normalized ITT only in 2-months for all groups?

A: At the beginning of the test, LFD2 and HFD2 mice show higher glucose values than C mice. However, 30 minutes after the administration of insulin intraperitoneally, these mice showed a decrease in glucose values, reaching similar values to C mice. The purpose of the ITT is to evaluate the action of the exogenous hormone in the living model [4–6]. It has been reported that the increase in insulin levels, a state of hyperinsulinemia, interferes with the internalization processes of the insulin receptor, favoring its increase in the cell membrane [7], and increased expression of the insulin receptor [9]. This can be activated and induce signaling pathways that favor glucose uptake, thus decreasing glucose concentration [9, 10]. However, the constant and chronic stimulation of caloric intake will inevitably lead to alterations in insulin sensitivity, as observed in the groups fed the diets for three months.

  1. Line: 455-456-“In the present work, oxidative damage at the brain level was evidenced by PCO increased (LFD2 mice) and decreased GSH-Px (LFD1 mice)”. What happens to the brain in 3 month group? What will be the possible reason for the normalized values of MDA, PCO and GSH-Px after 3 months of LFD?

A: The development of oxidative stress favors neuronal preconditioning, through the hormesis mechanism, (mechanism in which the cell responds to stressful situations to replenish its normal conditions) [9,10]. Additionally, it has been reported that the excessive intake of carbohydrates, can induce an increase in the production of ROS in cells [11], could triggers an adaptive response that promote the production of antioxidant enzymes, by Keap oxidation and Nrf2 activation. [12]. This result in a long-term adaptation process, inducing protective mechanisms, which could favors the reestablishment of neuronal functions, and its multiple functions in the brain [13]. Coincidentally, LFD3 and HFD3 animals also did not present BIR, since they responded again to the insulin stimulus in the POA, a situation that is evidenced by the increase in temperature detected by the devices implanted intraperitoneally.

  1. The discussion is too lengthy and describes too many aspects that are not even studied in the present study. The discussion part is more about the findings from other studies, rather than discussing the findings of the present by including previous studies. 

A: Thank you for your comment. We have summarized the discussion by highlighting the cause and importance of our results.

  1. Conclusion says that: “The use of HFD or LFD at different duration can influence the establishment of PIR or BIR, and oxidative damage in the fat tissue and cerebral cortex can play an important role”. However, the results of the study do not support the conclusion statement. FOR BIR, only 2 months showed an increase in %PCO in the cerebral cortex of LFD. Also, % of MDA was decreased in the cerebral cortex of LFD after 1 month and no significant differences were observed after subsequent months. With these results, the authors cannot conclude that oxidative stress plays an important role in LFD in BIR. More detailed studies are required to validate the aim.

A: Indeed, more studies are needed to clarify the specific role of oxidative stress in PIR or BIR. However, the alteration of oxidative damage markers such as MDA (marker of oxidative damage to lipids) and PCO (marker of oxidative damage to proteins) as the changes in the enzymatic activity of GPX suggest that oxidative damage is present at some stage in the development of PIR or BIR and may play an important role. Given that the increase in %PCO at two months in the LFD2 group and the decrease in GSH-Px activity (indicative of a deficiency in the activation mechanisms of the antioxidant system) in LFD1 coincides with the decrease in the thermogenic insulin response in these two groups LFD1 and LFD2 at specific times as shown in the figure 4 (4B y 4F), compared to their respective controls. Our results could define the first signs of oxidative damage in the cerebral cortex.

Indeed, at three months, the levels of these markers normalize, possibly due to cellular preconditioning. However, we do not rule out that chronic exposure to the diet could once again induce an increase in lipid and protein damage, as well as a prolonged decrease in enzymatic activity, as observed in several published studies [14–16].

The conclusions were revised for better clarity and understanding.

  1. Writing is incomprehensible with plenty of grammatical, typographical and punctuation errors.

A: Thank you for your comment. We will send the document for language correction.

  1. Overall, the study lack clarity in concluding the aim and looks scientifically weak.

A: Corrections have been made in the conclusion and the key points of the methodology have been referenced and what was done according to your comments has been substantiated.

It is worth mentioning that the in vivo model is an accepted and valid system for the study of metabolic alterations, because of its complexity due to the interaction between all tissues and organs; and the representativeness of what could occur in a human [2,17–19].

References

  1. Kobayashi, K.; Pillai, K.S. Repeated Dose Administration Toxicity Studies - Use of t-Tests in Multiplicity Data Analysis. Fundam. Toxicol. Sci. 2024, 11, 1–10, doi:10.2131/fts.11.1.
  2. Nagy, C.; Einwallner, E. Study of In Vivo Glucose Metabolism in High-Fat Diet-Fed Mice Using Oral Glucose Tolerance Test (OGTT) and Insulin Tolerance Test (ITT). JoVE 2018, 56672, doi:10.3791/56672-v.
  3. Czech, M.P. The Nature and Regulation of the Insulin Receptor: Structure and Function. Annu. Rev. Physiol. 1985, 47, 357–381, doi:10.1146/annurev.ph.47.030185.002041.
  4. Cózar-Castellano, I.; Perdomo, G. Assessment of Insulin Tolerance In Vivo in Mice. In Animal Models of Diabetes; King, A.J.F., Ed.; Methods in Molecular Biology; Springer US: New York, NY, 2020; Vol. 2128, pp. 217–224 ISBN 978-1-07-160384-0.
  5. Manwani, T.; Wanjari, A.; Acharya, S. Insulin Resistance and Its Detection. Journal of Datta Meghe Institute of Medical Sciences University 2024, 19, 416–420, doi:10.4103/jdmimsu.jdmimsu_171_23.
  6. Pedro, P.F.; Tsakmaki, A.; Bewick, G.A. The Glucose Tolerance Test in Mice. In Animal Models of Diabetes; King, A.J.F., Ed.; Methods in Molecular Biology; Springer US: New York, NY, 2020; Vol. 2128, pp. 207–216 ISBN 978-1-07-160384-0.
  7. Watson, L.S.; Wilken-Resman, B.; Williams, A.; DiLucia, S.; Sanchez, G.; McLeod, T.L.; Sims-Robinson, C. Hyperinsulinemia Alters Insulin Receptor Presentation and Internalization in Brain Microvascular Endothelial Cells. Diabetes and Vascular Disease Research 2022, 19, 147916412211186, doi:10.1177/14791641221118626.
  8. Copps, K.D.; White, M.F. Regulation of Insulin Sensitivity by Serine/Threonine Phosphorylation of Insulin Receptor Substrate Proteins IRS1 and IRS2. Diabetologia 2012, 55, 2565–2582, doi:10.1007/s00125-012-2644-8.
  9. Ristow, M.; Zarse, K. How Increased Oxidative Stress Promotes Longevity and Metabolic Health: The Concept of Mitochondrial Hormesis (Mitohormesis). Experimental Gerontology 2010, 45, 410–418, doi:10.1016/j.exger.2010.03.014.
  10. Sharma, V.; Mehdi, M.M. Oxidative Stress, Inflammation and Hormesis: The Role of Dietary and Lifestyle Modifications on Aging. Neurochemistry International 2023, 164, 105490, doi:10.1016/j.neuint.2023.105490.
  11. Tan, B.L.; Norhaizan, M.E.; Liew, W.-P.-P. Nutrients and Oxidative Stress: Friend or Foe? Oxidative Medicine and Cellular Longevity 2018, 2018, 9719584, doi:10.1155/2018/9719584.
  12. Dai, X.; Yan, X.; Wintergerst, K.A.; Cai, L.; Keller, B.B.; Tan, Y. Nrf2: Redox and Metabolic Regulator of Stem Cell State and Function. Trends in Molecular Medicine 2020, 26, 185–200, doi:10.1016/j.molmed.2019.09.007.
  13. Lee, K.H.; Cha, M.; Lee, B.H. Crosstalk between Neuron and Glial Cells in Oxidative Injury and Neuroprotection. IJMS 2021, 22, 13315, doi:10.3390/ijms222413315.
  14. Velloso, L.A.; Araújo, E.P.; De Souza, C.T. Diet-Induced Inflammation of the Hypothalamus in Obesity. Neuroimmunomodulation 2008, 15, 189–193, doi:10.1159/000153423.
  15. Cavaliere, G.; Trinchese, G.; Penna, E.; Cimmino, F.; Pirozzi, C.; Lama, A.; Annunziata, C.; Catapano, A.; Mattace Raso, G.; Meli, R.; et al. High-Fat Diet Induces Neuroinflammation and Mitochondrial Impairment in Mice Cerebral Cortex and Synaptic Fraction. Front. Cell. Neurosci. 2019, 13, 509, doi:10.3389/fncel.2019.00509.
  16. Hahm, J.R.; Jo, M.H.; Ullah, R.; Kim, M.W.; Kim, M.O. Metabolic Stress Alters Antioxidant Systems, Suppresses the Adiponectin Receptor 1 and Induces Alzheimer’s Like Pathology in Mice Brain. Cells 2020, 9, 249, doi:10.3390/cells9010249.
  17. Nedergaard, J.; Cannon, B. Diet-Induced Thermogenesis: Principles and Pitfalls. In Brown Adipose Tissue; Guertin, D.A., Wolfrum, C., Eds.; Methods in Molecular Biology; Springer US: New York, NY, 2022; Vol. 2448, pp. 177–202 ISBN 978-1-07-162086-1.
  18. Casimiro, I.; Stull, N.D.; Tersey, S.A.; Mirmira, R.G. Phenotypic Sexual Dimorphism in Response to Dietary Fat Manipulation in C57BL/6J Mice. Journal of Diabetes and its Complications 2021, 35, 107795, doi:10.1016/j.jdiacomp.2020.107795.
  19. Fraulob, J.C.; Ogg-Diamantino, R.; Fernandes-Santos, C.; Aguila, M.B.; Mandarim-de-Lacerda, C.A. A Mouse Model of Metabolic Syndrome: Insulin Resistance, Fatty Liver and Non-Alcoholic Fatty Pancreas Disease (NAFPD) in C57BL/6 Mice Fed a High Fat Diet. J. Clin. Biochem. Nutr. 2010, 46, 212–223, doi:10.3164/jcbn.09-83.

Round 2

Reviewer 2 Report

Comments and Suggestions for Authors

The authors have addressed the concerns and revised the manuscript appropriately.